# Intraurban socioeconomic inequalities in life expectancy: a population-based cross-sectional analysis in the city of Córdoba, Argentina (2015–2018)

Santiago Rodríguez López [1,2] Natalia Tumas [1,3] Usama Bilal [4,5]
Kari A Moore,[4] Binod Acharya,[4] Harrison Quick,[4,5] D Alex Quistberg [4,6]
Gabriel E Acevedo,[7] Ana V Diez Roux[4,5]

For numbered affiliations see end of article.

**Correspondence to**
Dr Santiago Rodríguez López; santiago.rodriguez@unc.edu.ar

## ABSTRACT

**Objectives** To evaluate variability in life expectancy at birth in small areas, describe the spatial pattern of life expectancy, and examine associations between small-area socioeconomic characteristics and life expectancy in a mid-sized city of a middle-income country.

**Design** Cross-sectional, using data from death registries (2015–2018) and socioeconomic characteristics data from the 2010 national population census.

**Participants/setting** 40 898 death records in 99 small areas of the city of Córdoba, Argentina. We summarised variability in life expectancy at birth by using the difference between the 90th and 10th percentile of the distribution of life expectancy across small areas (P90-P10 gap) and evaluated associations with small-area socioeconomic characteristics by calculating a Slope Index of Inequality in linear regression.

**Primary outcome** Life expectancy at birth.

**Results** The median life expectancy at birth was 80.3 years in women (P90-P10 gap=3.2 years) and 75.1 years in men (P90-P10 gap=4.6 years). We found higher life expectancies in the core and northwest parts of the city, especially among women. We found positive associations between life expectancy and better small-area socioeconomic characteristics, especially among men. Mean differences in life expectancy between the highest versus the lowest decile of area characteristics in men (women) were 3.03 (2.58), 3.52 (2.56) and 2.97 (2.31) years for % adults with high school education or above, % persons aged 15–17 attending school, and % households with water inside the dwelling, respectively. Lower values of % overcrowded households and unemployment rate were associated with longer life expectancy: mean differences comparing the lowest versus the highest decile were 3.03 and 2.73 in men and 2.57 and 2.34 years in women, respectively.

**Conclusion** Life expectancy is substantially heterogeneous and patterned by socioeconomic characteristics in a mid-sized city of a middle-income country, suggesting that small-area inequities in life expectancy are not limited to large cities or high-income countries.

## STRENGTHS AND LIMITATIONS OF THIS STUDY

⇒ The study had unique access to mortality data geo-referenced to small areas.

⇒ The Bayesian approach draws strength from surrounding small areas and from the overall structure of the mortality schedule allowing valid and reliable estimation even when data are sparse.

⇒ The use of the Slope Index of Inequality takes advantage of the full distribution of socioeconomic variables but avoids excessive influence of extreme values.

⇒ Excessive smoothing may have resulted in underestimated inequities.

⇒ Mortality and census population and socioeconomic data were not exactly aligned in time; thus, it was necessary to assume that the social characteristics of small areas, the population age, and sex structure were relatively stable across the years examined.

## INTRODUCTION

The social context of daily life is linked to health outcomes and generates social inequalities in health.[1] Health inequalities tend to be large in urban areas because cities include areas of concentrated deprivation and poverty,[2 3] especially in large cities where spatial and social inequalities are combined.[4] Latin America is one of the most urbanised regions of the world with urbanisation levels estimated to be in the order of 80%.[5] It is also one of the most unequal regions in the world and is home to more than half of the thirty most unequal cities in the world.[6]

Although a number of studies have documented large inequities in mortality and life expectancy across neighbourhoods within cities in high-income countries,[7–11] there has been very limited investigation of within-city heterogeneity in life expectancy in cities of lower income and middle-income countries. Indeed, some work has suggested that

the educational patterning of adult mortality in middle-income countries can be different from that observed in high-income countries.[12]

Argentina, one of the most urbanised countries in Latin America, is a middle-income country with large social and health disparities.[13] However, evidence of intraurban health variations across small areas is still scarce. Prior research showed that higher income inequality at the provincial level was associated with lower life expectancy for both men and women.[14] Another study described significant associations between social inequity and premature death rates in departments of Argentina.[15] Diez Roux et al[4] described substantial intraurban variation in the risk of death in the core area of the large urban agglomeration of Buenos Aires, finding similar patterns to another study examining life expectancy across the entire metropolitan area.[16] These studies are primarily focused on the city of Buenos Aires, but no studies of which we are aware have so far addressed variability and the social patterning of life expectancy at birth in smaller cities which may be more representative of a larger number of cities across the Latin American region.

The city of Córdoba, Argentina, provincial capital of 1.3 million residents, is emblematic of emerging cities across the region as a medium-sized city with fast urban growth. As is the case for other cities, generating local evidence to fill the knowledge gap on existing social inequities in life expectancy at birth in the city of Córdoba is relevant to informing policies to improve health conditions.[17] Examining these issues in Córdoba can both inform local policy while also provide a case study of an emerging medium-sized city in the region. Prior research has suggested substantial heterogeneity of life expectancy and mortality in Latin American cities,[16 18] thus exploring these issues in small areas across heterogeneous cities is important for understanding the need to address them in different urban contexts and development levels, especially as much of the future urban growth in the region will likely be in cities like Córdoba.[19]

To assess the extent to which spatial patterning of life expectancy and its association with small-area socioeconomic characteristics extends beyond the largest urban agglomerations to other cities in middle-income countries, we examined small-area variations in life expectancy in the city of Córdoba. Our goals were to (1) evaluate the variability in life expectancy at birth in small areas of the city of Córdoba for the 2015–2018 period; (2) describe the spatial patterning of life expectancy at birth and; (3) examine the association between small area socioeconomic characteristics and life expectancy at birth. We hypothesised that (1) there is considerable variability of life expectancy at birth across small areas of Córdoba; (2) there is lower life expectancy in the periphery of the city of Córdoba as compared with core areas; and (3) small areas with better socioeconomic characteristics have a higher life expectancy at birth.

## METHODS

### Study setting
We used data compiled by the SALURBAL project (Urban Health in Latin America; 'Salud Urbana en América Latina') for the city of Córdoba during the years 2015–2018, which has compiled and harmonised data on health, social and built environment indicators for a large number of cities in different Latin American countries.[20] The city of Córdoba (Argentina), the capital and main city of Córdoba province, is a highly urbanised area of 576 km$^2$ and around 1.3 million residents. We defined small areas as the 99 *fracciones censales* (spatially delimited units used for census data collection). These small areas had a median population (5th–95th percentile) of 13 370 (5837–22 031) individuals in 2010 (median annual average for the 2015–2018 period: 14 389 (6305–23 567)).

### Data sources
We used data from (1) death registries georeferenced to small areas during the years 2015–2018, (2) projected population estimates by age, sex and municipality for 2015–2018, and (3) population and socioeconomic characteristics by small area from the 2010 national population census of Argentina.[21]

### Mortality data
Decedents' residential addresses were extracted from death certificates provided by the Civil Registry of the City of Córdoba, and were georeferenced to latitude and longitude coordinates using the ESRI World Geocoding Service by the SALURBAL team. We used all mortality records for the city of Córdoba from 2015 to 2018 (n=42 115) that included residential addresses of the deceased. We georeferenced these records to one of 99 small areas, and excluded records georeferenced to areas outside of the administrative boundaries of Córdoba (n=1084) and to non-residential points of interest (n=133). Of the remaining 40 898 georeferenced death records, n=4075 (~10%) had low georeferencing accuracy (ie, matched to street name but not building number) but could still be assigned to a small area using the centre point of the street or area they were georeferenced to, and were included in the study.

### Population data
First, we obtained population denominators by sex, single years of age (0 to 110+years), and small areas from 2010 Argentina census and calculated the proportion of people in the city of Córdoba by age, sex, and small area. Second, we obtained projected population estimates provided by the National Institute of Statistics and Census of Argentina for 2015–2018 by 5 year age groups (0 to 85+years) and sex for the whole municipality of Córdoba and graduated these data into single ages using a penalised composite link model.[22] We then obtained small-area population by single age and sex for 2015–2018 by applying the 2010 census proportions to the graduated municipality-level population (2015–2018).

## Socioeconomic characteristics of small areas

We used the following variables to proxy socioeconomic characteristics of small areas: % of the population aged 15–17 years attending school, % of the population aged 25 years or above with completed secondary education or above, % of households with piped water access inside the dwelling, % households with overcrowding (defined as more than three people per room), and unemployment rate among individuals aged 15 years and above. Area-level education has been associated with variations in intraurban mortality in Argentina,[4 16] and water access, sanitation, and less overcrowding[18] have been also associated with higher life expectancy in Latin American cities. In addition, and in order to capture broader features of the social environment, we created a composite z-score index combining the variables listed above. Although a multidimensional index only captures some aspects of the urban infrastructure and resources,[18] it is likely a better proxy of the social environment than a single indicator. Combined indices including education indicators have been frequently used in health disparities research.[23 24] Before creating the composite z-score index, we standardised each of these variables (unemployment and overcrowding were reversed by multiplying with –1). The average of these standardised scores was defined as the composite z-score representing the socioeconomic characteristics of the small areas.

## Statistical analysis

To estimate life expectancy, we need an estimate of age-specific, sex-specific, and small-area-specific mortality rates. We estimated these through a Bayesian adaptation of the TOPALS (tool for projecting age patterns using linear splines method), incorporating spatial smoothing between small areas.[25] TOPALS regression requires the use of a standard mortality schedule. The core idea is that the schedule of log mortality rates to be estimated is the sum of the standard schedule and a linear spline function. Following the original TOPALS method,[25] we used the city-level Córdoba mortality schedule as the standard schedule. Other research has shown that the choice of the standard mortality schedule does not fundamentally alter results.[26] Because the log mortality rates of the standard mortality schedule were noisy (ie, contained unexplained variance) (see online supplemental figure 1), we fit a LOESS regression, and the resulting smoothed rates were used as the standard mortality schedule. The models were run in WinBUGS[27] by using the R package R2Win-BUGS[28] for 100 000 iterations and the first 80 000 samples were discarded as *burn-ins*. The remaining samples were thinned by a factor of 10 to reduce the autocorrelation of the samples.[18] We fit the models for women and men separately. Eventually, we retained 2000 sets of age-specific, sex-specific, and small-area-specific mortality rates from the posterior distribution. See the online supplemental material for further details on how mortality rates were modelled.

To calculate life expectancy at birth, we inputted age-specific and sex-specific mortality rates into single-age life tables using the DemoTools package in R.[29] Life expectancy at birth is a simple indicator to present differences in mortality across and within populations,[30] and is defined as the number of years someone born today is expected to live if current age-specific mortality patterns hold constant in the future. These life tables were calculated for each of the 2000 sets of mortality rates resulting in 2000 sex-specific and small-area-specific life expectancy at birth estimates. For descriptive purposes, we report a point estimate (median) and 95% credible intervals (2.5th and 97.5th percentiles). We also extracted life expectancy at ages 20, 40 and 60 years from life tables to explore what ages could drive differences in life expectancy at birth.

To estimate the amount of variability in life expectancy at birth for each sex (hypothesis 1), we calculated the difference between 90th and the 10th percentile of the distribution of life expectancy at birth (P90-P10 gap) across small areas. The P90-P10 gap represents the variability in life expectancy across the city. To analyse the geographical patterning of life expectancy at birth (hypothesis 2), we presented choropleth maps of life expectancy at birth for men and women in small areas using ArcGIS. Finally, to examine the association between life expectancy at birth and socioeconomic characteristics of small areas (hypothesis 3), we fit univariate linear regressions of life expectancy on each predictor variable converted into deciles and scored on a continuous scale between 0 to 1. Specifically, for each socioeconomic variable, we assigned the value of 0 if it corresponds to the first decile of its distribution across all small areas. The second decile got the value of 1/9, the third decile got the value of 2/9, and so on. To acknowledge uncertainty around the estimates of life expectancy, these linear regressions were repeated 2000 times, one per life expectancy estimate. We used Rubin's formula to pool coefficients to obtain a single regression coefficient and associated standard errors. Each coefficient represents the mean difference in life expectancy in areas with the highest socioeconomic variable (ie, those in the tenth decile) versus the areas with the lowest value of the socioeconomic variable (those in the first decile), and is presented as the Slope Index of Inequality (SII). Linear regression models were run using PROC REG and PROC MIANALYZE in SAS software.

## Patient and public involvement

SALURBAL has launched a series of Knowledge-to-Policy Fora to present preliminary results and engage urban health policy actors from across Latin America in dialogue on urban health research and policy priorities in the region. For more information, see https://drexel.edu/lac/events-workshops/knowledge-policy-forum/. The project also holds and participates in additional periodic workshops to engage stakeholders in activities designed to disseminate research findings and engage stakeholders in systems thinking around urban policies.

**Table 1** Number of deaths, life expectancy among women and men, and selected sociodemographic characteristics for small areas (n=99). Córdoba, 2015–2018

| | Median (10th–90th percentile) | |
| --- | --- | --- |
| | **Women** | **Men** |
| Number of deaths | 222 (90, 340) | 199 (81, 309) |
| Estimated population, annual average* | 7326 (3,818, 11,157) | 6874 (3,214, 10,534) |
| Life expectancy at birth, years* | 80.3 (78.3, 81.5) | 75.1 (72.5, 77.1) |
| Life expectancy at age 20, years* | 61.0 (59.1, 62.2) | 56.2 (53.6, 57.9) |
| Life expectancy at age 40, years* | 41.5 (39.8, 42.6) | 36.9 (34.7, 38.6) |
| Life expectancy at age 60, years* | 23.3 (21.6, 24.3) | 19.2 (17.2, 21.0) |
| Population aged 15 or younger, %* | 22.0 (8.7, 30.3) | 25.1 (10.6, 32.2) |
| Population aged 65 or older, %* | 13.3 (7.1, 19.5) | 8.9 (5.3, 12.5) |
| Households with water inside dwelling, % † | 97.6 (89.8, 99.4) | |
| Households with overcrowding, % † | 2.1 (0.4, 7.1) | |
| School attendance among 15–17 years old, % † | 85.3 (76.2, 91.9) | |
| Adults with high school education or above, %† | 58.4 (30.1, 88.7) | |
| Unemployment, %† | 7.0 (4.6, 9.7) | |

*Data from 2015 to 2018.
†Data from 2010 census. Overcrowding: proportion of households with more than three people per room.

## RESULTS

A median of 222 and 199 deaths occurred during the study period per small area among women and men, respectively (table 1). A median of 85.3% of adolescents between 15 and 17 years attended school, 58.4% of people had minimum high school education, and 7.0% were unemployed (table 1). Moreover, 97.6% of dwellings had inside water connections and 2.1% of households experienced overcrowding. Median life expectancy at birth was 80.3 years for women with variability in life expectancy (P90-P10 gap) of 3.2 years (P90=81.5 years and P10=78.3 years) (table 1). Among men, there was lower median life expectancy at birth (75.1 years), with a gap (P90-P10) of 4.6 years between small areas (P90=77.1 years, and P10=72.5 years). For the full distribution of life expectancy at birth and uncertainty around the estimates, see online supplemental figure S2.

There were higher life expectancies in the core and northwestern parts of the city for both men and women, along with the southwestern part of the city for men (figure 1). The distribution of life expectancy at ages 20, 40 and 60 years showed similar spatial patterns to those for life expectancy at birth in women and men (see online supplemental figure S3).

Overall, there were positive associations between life expectancy at birth and better socioeconomic characteristics of small areas (figure 2, table 2), with associations being slightly stronger among men. Living in small areas with the highest compared with the lowest decile of high school education and school attendance was associated with longer life expectancy (SII=3.03 and 3.52 years, respectively, in men and 2.58 and 2.56 years, respectively, in women) (table 2). Similarly, higher proportions of

households with water inside the dwelling was associated with longer life expectancy by 2.97 years in men, and 2.31 years in women, while lower overcrowding and % unemployment was associated with shorter life expectancy at birth by 3.03 and 2.73 years, respectively, and 2.57 and 2.34 years for women (table 2). The composite z-score was also strongly associated with life expectancy, as higher values (indicating better living conditions) were associated with 3.27 and 2.69 years longer life expectancy in men and women, respectively.

## DISCUSSION

Our study evaluated the variability in life expectancy at birth, described the spatial patterning of life expectancy, and examined the association between life expectancy and socioeconomic characteristics of small areas in the city of Córdoba (Argentina) during the 2015–2018 period. We found a higher life expectancy for women (5.2 years higher than for men), that life expectancy varied within the city of Córdoba, with a P90-P10 gap of 4.6 years for men and 3.2 years for women, and that life expectancy was higher in the central and northwestern parts of the city, along with the southwestern part in the case of men. Lastly, we found that a series of small-area socioeconomic characteristics were highly predictive of life expectancy.

Analyses of spatial variability provide critical information for understanding place effects on health.[31] We found considerable variability (P90-P10 gap) in life expectancy at birth across small areas of Córdoba, with this gap being larger in men (4.6 years) than in women (3.2 years). In another study, we found that variability in rates of non-communicable disease risk factors was

 Rodríguez López S, *et al. BMJ Open* 2022;**12**:e061277. doi:10.1136/bmjopen-2022-061277

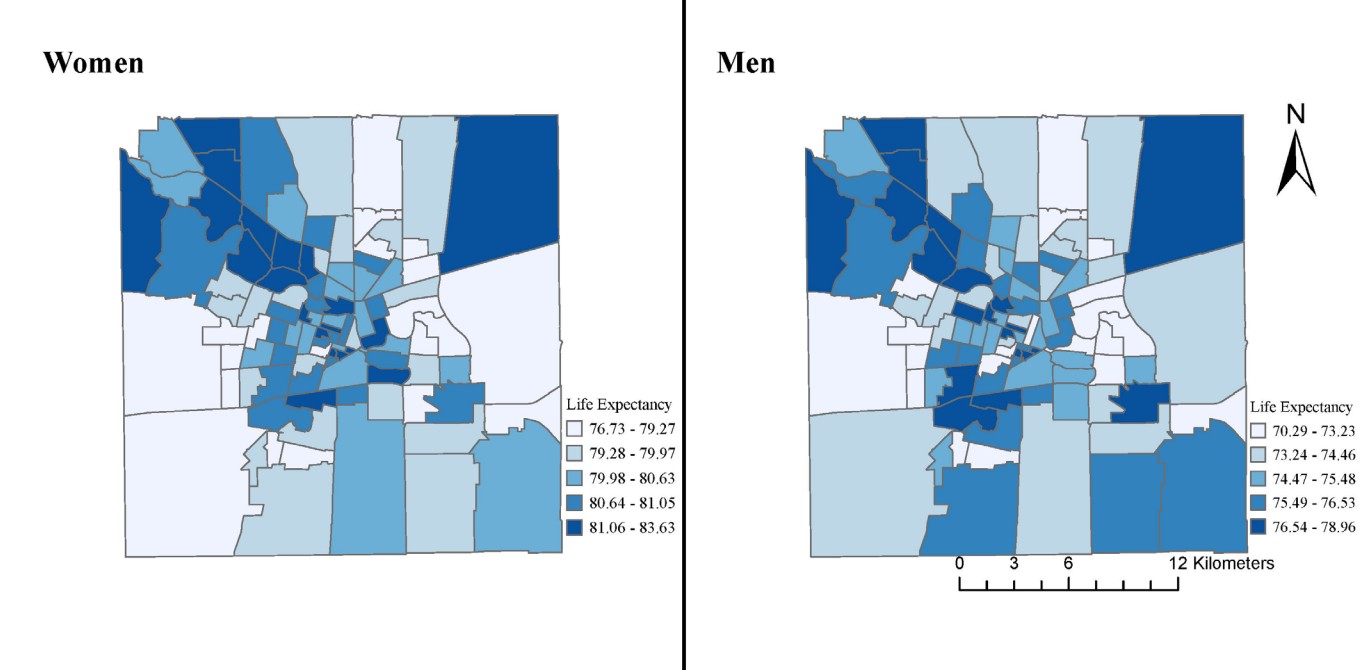

**Figure 1** Spatial distribution of life expectancy at birth in women and men in the city of Córdoba, 2015–2018. All small areas (n=99) of the city are indicated. To facilitate within-sex comparisons, each map uses its own legend.

larger across neighbourhoods within cities than across cities,[32] further highlighting the importance of within-city inequities. A recent study on the variations in life expectancy across states, counties, and census tracts in the US showed that more than three-fourths of the total variation in life expectancy was attributable to census tract units, suggesting that large heterogeneities in longevity occur at a local level.[9]

Some prior work has demonstrated within-city heterogeneities in life expectancy in lower and middle-income countries. A prior SALURBAL paper based on six large cities in the region found that gaps in life expectancy across subcity units ranged from a high of 15.0 years in men from Panama City (Panamá) to a low of 3.0 years in women from San José (Costa Rica).[16] However, this analysis focused on only the largest cities and also investigated subcity areas of varying size but all larger than the ones we investigated in Córdoba. Very few studies examined variation across smaller areas and most of these were in large cities. A study in Rio de Janeiro[33] showed that life expectancy at birth of men living in the richest parts of the city was 12.8 years longer than that of men living in deprived areas. In Argentina, one prior study focused on the city of Buenos Aires demonstrated large variations in mortality rates across censal fractions but life expectancy differences were not explored.[4]

Comparison across studies in the size of within-city differences is rendered complex by the use of geographic units of different sizes and the way in which the city is defined. For example, some articles[16] have used larger sub city geographical units (which could perhaps hide heterogeneity) but have also employed a broader definition of cities, encompassing peripheral areas around the core city that are often poorer (possibly leading to the larger estimates of within-city differences). The metrics used to characterise within-urban differences also vary. We compared the 90th to the 10th percentile to avoid excessive influence of extreme values but others compare maximum and minimum.[7 11] On a more substantive level, the degree of residential segregation may also impact the magnitude of heterogeneity in life expectancy across areas within a city.[11] Further work is needed to contrast and understand the drivers of differences across cities in the size of small-area differences in life expectancy.

In addition to documenting large heterogeneity across areas, we also found that life expectancy at birth was geographically patterned, with higher life expectancy in the core and northwestern parts of the city, for both men and women, along with the southwestern part of the city for men. This suggested that variability in life expectancy was not random in Córdoba. Indeed, the city of Córdoba has an important socioeconomic residential segregation[34] that matched the spatial pattern of life expectancy: while the north-east-south periphery of the city predominately includes residential areas with poverty segregation, the central core, and the northwest corridor are characterised by a greater concentration of wealth. There has been growing spatial concentration and social isolation of households with a lower socioeconomic position as a consequence of public housing policies implemented in recent decades in the city of Córdoba. Particularly, relocations from informal settlements to large housing complexes built in the poor and semirural periphery of the city have been implemented by the provincial government between 2003 and 2007.[34 35] The spatial pattern in Córdoba, which

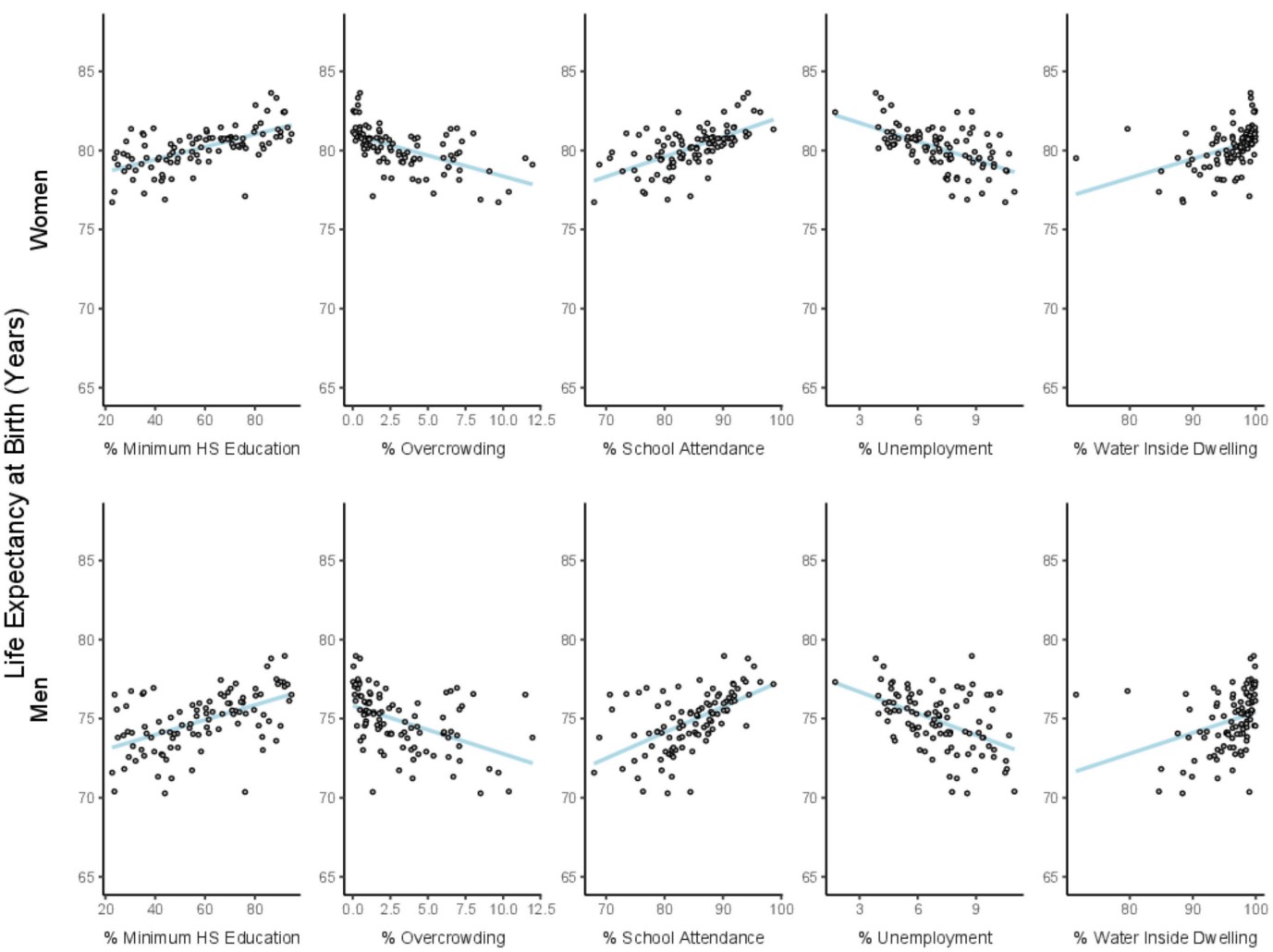

**Figure 2** Association between life expectancy at birth and socioeconomic characteristics of small areas in women and men. Córdoba, 2015–2018. Blue line refers to the linear fit.

exhibits the highest life expectancy in the core and the northwest, was more clearly defined for women, while we also found higher life expectancy in the southern periphery among men. Previous research showed a similar core-periphery divide[16] and a north-to-south pattern in Buenos Aires, which can be explained to some extent by the spatial patterning of socioeconomic deprivation. Our findings emphasise the need to address

**Table 2** Slope Index of Inequality (SII) in life expectancy (years) associated with small area characteristics among women and men, Córdoba city, 2015–2018

| Variable/model | Slope Index of Inequality (95% CIs) | |
| --- | --- | --- |
| | Women | Men |
| % with High school education or above | 2.58 (1.63 to 3.52) | 3.03 (1.77 to 4.30) |
| % School attendance among adolescents (15–17 years) | 2.56 (1.65 to 3.47) | 3.52 (2.35 to 4.69) |
| % households with water inside dwelling | 2.31 (1.36 to 3.26) | 2.97 (1.72 to 4.22) |
| % households with overcrowding | −2.57 (−3.51 to 1.63) | −3.03 (−4.29 to 1.77) |
| % Unemployment | −2.34 (−3.27 to 1.41) | −2.73 (−3.98 to 1.49) |
| Composite z-score | 2.69 (1.78 to 3.60) | 3.27 (2.05 to 4.48) |

The models were run in a univariate fashion, one variable at a time. Small-area characteristics were transformed into deciles. The Slope Index of Inequality (SII) represents the mean difference in life expectancy in areas with the highest predictor variable (ie, those in the tenth decile, having value=1) versus the areas with the lowest value of the predictor variable (those in the first decile, having value=0) as estimated from a linear regression including all deciles. Socioeconomic data for small areas came from the 2010 census. Overcrowding: proportion of households with more than three people per room.

spatial variation in health within cities in further studies in the local context.

We also explored factors that might explain the heterogeneity in life expectancy across small areas. We found positive associations between improved socioeconomic characteristics of small areas and life expectancy at birth in both men and women, suggesting that the described heterogeneity in life expectancy might be driven by the socioeconomic features of small areas. All factors including educational attainment, school attendance, water access, overcrowding, unemployment and the composite index were significantly associated with life expectancy at birth in the expected direction. These associations were stronger among men, a finding that could be driven by certain causes of death and that deserves further exploration. Our findings are also consistent with other research in Argentina describing a negative association between an area-level deprivation index and life expectancy at birth in 32 urban agglomerations,[24] and a positive association between mortality rate and unmeet basic needs in Departamentos of the 'Pampeana' region.[36] Furthermore, these results are consistent with factors associated with within-city inequalities in life expectancy in six large Latin American cities.[16]

A key strength of our analyses is that we employed a novel Bayesian approach that draws strength and smooths estimates from surrounding small areas and from the overall structure of the mortality schedule. This allowed us to derive valid and reliable estimates for small areas even in the presence of sparse data.[37] We also characterised associations with socioeconomic variables using the SII, a measure that uses the full distribution to derive a global measure of inequities. By using the SII based on deciles, we avoided the undue influence of extreme values of socioeconomic factors. The use of the SII defined in this way will also allow easier comparisons across cities. While some Latin American countries have vital registration records with incomplete registration, this is less of a concern in Argentina.[18 38 39] Another strength is that we examined these associations using univariate models that acknowledge uncertainty around the estimates of life expectancy by applying multiple imputation type techniques.

Among the limitations of our study was the relatively high proportion of records that had low georeferencing accuracy, that is, that were georeferenced to the street level. It was likely that these records were not randomly distributed. In fact, low georeferencing accuracy seemed to impact mostly small areas in the periphery. Given the challenges of sparse data, we also needed to apply spatial and age smoothing to address the small-area estimation of mortality rates. This smoothing could have led to an underestimation of inequalities, as reported in previous research.[16] Because annual small-area population by age and sex were not available, we had to estimate these by combining official municipality-level projected population estimates from 2015 to 2018 with the most recent census in 2010. If the population age or sex structure changed over time differentially by area, our estimates may be biased. Relatedly, social characteristics of small areas were retrieved from the same 2010 census and do not align with the years for which death records were obtained (2015–2018). We, therefore, assumed that social characteristics were relatively stable across the years examined. While ideally this analysis would have used census data from other years to better assess small-area changes over time, we could only rely on the 2010 census because small-area data from the 2001 Argentina census were not available, and the 2020 census was postponed to 2022 due to the COVID-19 pandemic, and data are not available yet. We explored life expectancy as it is a commonly used metric but other metrics such as age at death may also be of value to examine.

## Conclusion

In summary, we find that spatial variations in life expectancy across small areas within cities linked to small-area social and economic conditions extend beyond high-income countries and beyond the very large metropolitan areas in which they have been predominantly studied. These within-cities heterogeneity that are likely present or emerging in many small-sized and middle-sized cities all over the world are of major relevance for urban policies to promote health and achieve health equity in low-income and middle-income countries.

**Author affiliations**
[1]Centro de Investigaciones y Estudios sobre Cultura y Sociedad, Consejo Nacional de Investigaciones Científicas y Técnicas, Córdoba, Argentina
[2]Cátedra de Antropología, Departamento de Fisiología, Facultad de Ciencias Exactas Físicas y Naturales, Universidad Nacional de Córdoba, Córdoba, Argentina
[3]Research Group on Health Inequalities, Environment, and Employment Conditions Network (GREDS-EMCONET), Department of Social and Political Science, Universitat Pompeu Fabra, Barcelona, Spain
[4]Urban Health Collaborative, Dornsife School of Public Health, Drexel University, Philadelphia, Pennsylvania, USA
[5]Department of Epidemiology and Biostatistics, Dornsife School of Public Health, Drexel University, Philadelphia, Pennsylvania, USA
[6]Department of Environmental and Occupational Health, Dornsife School of Public Health, Drexel University, Philadelphia, Pennsylvania, USA
[7]Cátedra de Medicina Preventiva y Social, Facultad de Ciencias Médicas, Universidad Nacional de Córdoba, Córdoba, Argentina

**Acknowledgements** The authors acknowledge the contribution of all SALURBAL project team members. For more information on SALURBAL and to see a full list of investigators, see https://drexel.edu/lac/salurbal/team/. We would like to thank Steve Melly for his help in creating the maps for this study. We thank Eugenia Peisino and Alicia Díaz for their help in accessing to the mortality data.

**Contributors** SRL, NT, UB, KAM and AVDR conceived the study. BA and HQ did the statistical analyses. SRL, NT and BA drafted the first version of the manuscript. SRL, NT, KAM, ADQ, GEA and AVDR participated in or supported data collection. AVDR is responsible for the overall content as guarantor. All authors participated in the interpretation of the results and approved the final version of the manuscript.

**Funding** The Salud Urbana en América Latina (SALURBAL)/ Urban Health in Latin America project is funded by the Wellcome Trust [205177/Z/16/Z]. NT was awarded by the European Union's Horizon 2020 research and innovation programme with the Marie Sklodowska-Curie grant agreement No. 89102. UB was supported by the Office of the Director of the National Institutes of Health under award number DP5OD26429.

**Competing interests** None declared.

**Patient and public involvement** Patients and/or the public were involved in the design, or conduct, or reporting, or dissemination plans of this research. Refer to the Methods section for further details.

**Patient consent for publication** Not applicable.

**Ethics approval** The SALURBAL study protocol was approved by the Drexel University Institutional Review Board with ID #1612005035 and by appropriate site-specific IRBs.

**ORCID iDs**
Santiago Rodríguez López http://orcid.org/0000-0002-3539-3751
Natalia Tumas http://orcid.org/0000-0003-4730-6624
Usama Bilal http://orcid.org/0000-0002-9868-7773
D Alex Quistberg http://orcid.org/0000-0001-9730-2686

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
