## [Reviewer comments · BMJ Open]

ARTICLE DETAILS

TITLE (PROVISIONAL)	Intraurban socioeconomic inequalities in life expectancy: a population-based cross-sectional analysis in the city of Córdoba, Argentina (2015-2018)
AUTHORS	Rodríguez López, Santiago; Tumas, Natalia; Bilal, Usama; Moore, Kari; Acharya, Binod; Quick, Harrison; Quistberg, Alex; Acevedo, Gabriel; Diez Roux, Ana V.

VERSION 1 – REVIEW

REVIEWER	Canudas-Romo, Vladimir Australian National University, School of Demography, College of Arts and Social Sciences
REVIEW RETURNED	08-Mar-2022

GENERAL COMMENTS	Review Life expectancy in small areas of the city of Cordoba, Argentina This publication is part of the wider project of the SALURBAL. The data of that project is still not available, so it is becoming worrisome that readers and reviewers do not have actual access to the information. I know this because I reviewed positively a paper on this published in nature medicine, but I was expecting that then the data would become available for all the research community, which still is not the case. This particular manuscript is missing justification. Why should I learn about Cordoba? The explanation given is that the manuscript is not about Buenos Aires. That is not enough. I would have much preferred to see comparison between cities in the country, particular if also small cities could be included in the analysis. Here below specific pages and questions that I had while reading your text (I repeat your text as “ and my comments followed by a -) --- Page 4 --- “P90-P10 gap” - What is p10 or p90? Clarify in the abstract or leave out Not clear what are those “mean differences in life expectancy in years per SD higher value...” - Not clear what is that calculation for? “acroos”
---

--- Page 5 ---

“These studies are primarily focused on the city of Buenos Aires, but no studies of which we are aware have so far addressed variability and the social patterning of life expectancy at birth in Córdoba, the second most populated city in the country.”

- Why should this be of interest to an international audience? I found this explanation on why looking at Córdoba rather poor. More justification is needed. I would think that comparing BA vs Córdoba as a within country comparison would be much more interesting and appealing for the general reader of BMJ Open

“1) evaluated the variability in life expectancy at birth

- Why not the variability at age at death? Which is actually something of current interest among researchers of mortality

“We hypothesized that (1) there is considerable variability of life expectancy at birth across small areas of Córdoba; (2) there is lower life expectancy in the periphery of the city of Córdoba as compared to core areas; and

- Not enough justification of why is this important beyond for those researchers studying Córdoba

“3) small areas with better socioeconomic characteristics have higher life expectancy at birth.

- That seems obvious, the rich are living longer all over the world, or what is the research question?

--- Page 6 ---

“death registries geocoded to small areas during the years 2015-2018

- Details about this death information is important to understand if the allocation of the deaths actually correspond to the living place or the area where hospitals are found

“geocoded

- What’s that and how was it done? The place of usual living for the deaths?

“we we

“Second, we obtained population projections

- From where?

“We used all mortality records for the city of Cordoba from 2015 to 2018

- Where is the information coming from?

“socioeconomic characteristics by small area from the 2010 national population census

	- It is extremely inconvenient that the timing of the data is not matching. --- Page 7 --- “standard mortality schedule were noisy - Noisy? Doesn’t sound as a way of describing rates in a scientific journal “calling the software “discarded as burn-ins “samples were thinned - A lot of jargon. Doesn’t sound as usual wording for a methods section of a manuscript. “To estimate the amount of variability in life expectancy at birth for each sex (hypothesis 1), we calculated the difference between 90th and the 10th percentile of the distribution of life expectancy at birth (P90-P10 gap) - that needs to be clarified in the abstract --- Page 8 --- “A median of 222 and 199 deaths occurred during the study period per small area among women and men, respectively (Table 1). - Those are very small numbers of deaths which if proper confidence intervals had been calculated will translate in very big CIs. Then how to compare the P90 to the P10 and their CIs, for example if the CIs overlap. Not clear. “The distribution of life expectancy at ages 20, 40, and 60 years, showed similar patterns to those for life expectancy at birth in women and men (see Figure S3, Supplementary material). - Life expectancy by age is practically a straight line and you are looking at parallel lines when comparing the different small areas in the city. --- Page 9 --- “1-SD - Not clear why first you show some results with P90-P10 and then variability using 1-SD. --- Page 10 --- “Argentina has been mainly focused in the city of Buenos Aires.[4,26] - Again, then why not a comparison between the main cities of Argentina
--	---

	--- Page 12 --- “Their distribution showed similar patterns to those reported for life expectancy at birth, suggesting that our estimates of life expectancy at birth are reliable and robust to those factors. - The strong relation between mortality levels at different ages is at the core of empirical model life tables. So your finding that results match across ages is not a reflection of better or worse estimation it is simply a result of this strong mortality relations at different ages. “We also had to combine population projections - What do you mean by combining population projections? And do you mean population projections? That is usually into the future “If the population age or sex structure changed over time differentially by area, our estimates may be biased - This mismatching in the timing of the information should be somehow incorporated in the regression analysis. For example, different census (2000 and 2010, 2020) could be used to actually confirm such fixed values or somehow do interpolation of the information. --- Page 17 --- Figure 1 - How is density included in the calculations of life expectancy? --- Page 18 --- Figure 2 - Strange, confidence intervals do not cover the values, and does not align to them. In other words, I would expect that if there was greater number of observed dots with greater disparity that the CI will cover the great majority... that is not the case, instead giving wide CI when single dots are found - What are the yellow and blue lines within CI? --- Page 20 --- Figure S1 - Why do you need to smooth them? Justification is needed --- Page 21 --- Figure S2 - What is the vertical axis? Label?
--	--

REVIEWER	Andreozzi, Lucia
REVIEW RETURNED	21-May-2022

GENERAL COMMENTS	I consent to the publication of this review, it is clear, well written and it uses statistical models properly.
---

VERSION 1 – AUTHOR RESPONSE

REVIEWER 1

Dr. Vladimir Canudas-Romo, Australian National University

Comments to the Author:

Review

Life expectancy in small areas of the city of Cordoba, Argentina

Comment 1.1

This publication is part of the wider project of the SALURBAL. The data of that project is still not available, so it is becoming worrisome that readers and reviewers do not have actual access to the information. I know this because I reviewed positively a paper on this published in nature medicine, but I was expecting that then the data would become available for all the research community, which still is not the case.

Response: Thank you for highlighting this. Unfortunately, the data use agreement we have signed with the city of Cordoba (Argentina) does not allow us to share the vital registration data at the small area level openly. Interested parties could submit a proposal to use the data and sign a data use agreement if approved. For the rest of the data, we are preparing an open data platform as required by the funder of the SALURBAL project, Wellcome Trust. For any other data requests, we include the following data sharing statement at the end of the manuscript:

“Interested parties could submit a proposal to SALURBAL to use the vital registration data at the small area level and sign a data use agreement if approved. The SALURBAL project welcomes queries from anyone interested in learning more about its dataset and potential access to data. To learn more about SALURBAL’s dataset, visit <https://drexel.edu/lac/> or contact the project at salurbal@drexel.edu.”

Comment 1.2

This particular manuscript is missing justification. Why should I learn about Cordoba? The explanation given is that the manuscript is not about Buenos Aires. That is not enough. I would have much preferred to see comparison between cities in the country, particular if also small cities could be included in the analysis.

Response: We appreciate this concern. We would argue that due to the lack of small area studies on health, mortality, and life expectancy in any cities in the region that this study will be of interest to many readers, not only within Argentina and the region, but also in other low- and middle-income countries. Córdoba, as we describe in the text we have added to the manuscript, is emblematic of emerging cities in the region as a medium-sized city with fast urban growth. Particularly as much of the future urban growth in the region will be in cities like Córdoba (see Kii 2021; <https://www.nature.com/articles/s42949-020-00007-5>; <https://www.oecd.org/cfe/regionaldevelopment/future-cities.htm>). Understanding small-area health inequalities in these cities is key to addressing current and preventing future urban health issues. We have added the following text to the introduction:

“Although a number of studies have documented large inequities in mortality and life expectancy across neighborhoods within cities in high-income countries,[7–11] there has been very limited investigation of within-city heterogeneity in life expectancy in cities of lower- and

middle-income countries. Indeed, some work has suggested that the educational patterning of adult mortality in middle-income countries can be different from that observed in high-income countries.[12]

“These studies are primarily focused on the city of Buenos Aires, but no studies of which we are aware have so far addressed variability and the social patterning of life expectancy at birth in smaller cities which may be more representative of a larger number of cities across the Latin American region.”

“The city of Córdoba, Argentina, provincial capital of 1.3 million residents, is emblematic of emerging cities across the region as a medium-sized city with fast urban growth. As is the case for other cities, generating local evidence to fill the knowledge gap on existing social inequities in life expectancy at birth in the city of Córdoba is relevant to informing policies to improve health conditions.[17] Examining these issues in Córdoba can both inform local policy while also provide a case study of an emerging medium-sized city in the region. Prior research has suggested substantial heterogeneity of life expectancy and mortality in Latin American cities[16,18], thus exploring these issues in small areas across heterogeneous cities is important for understanding the need to address them in different urban contexts and development levels, especially as much of the future urban growth in the region will likely be in cities like Córdoba.[19]

“To assess the extent to which spatial patterning of life expectancy and its association with small-area socioeconomic characteristics extends beyond the largest urban agglomerations to other cities in middle-income countries, we examined small-area variations in life expectancy in the city of Córdoba.”

The rationale for doing this single-city research is to i) fill the lack of local knowledge in the second largest city of Argentina, which we believe is important *per se*, in providing an advocacy tool to raise awareness about the issue and inform local policymakers, and ii) add to the scarce evidence on this type of studies in cities of Latin America, which are broadly characterized by high levels of deprivation and inequality. The latter is particularly relevant since extant research indicates that area-level characteristics might have a differential association with mortality by country (Bilal et al., 2021) and that adult mortality and risk factor gradients in middle-income countries can be much different than the established patterns seen in high-income countries (Sudharsanan et al., 2020). In other words, the specific social gradient observed in high income countries cannot be taken for granted and should be examined in other settings.

Nevertheless, we understand that a single-city study might not be as attractive as a comparative multi-city study within Argentina. Unfortunately, there is currently a lack of data on the distribution of socioeconomic characteristics and high-quality georeferenced deaths in small areas in Argentina, even more so when it comes to smaller cities. To the authors' knowledge, this type of research is (currently) being conducted only for the city of Buenos Aires and no other study has been published in any big or small city in Argentina. Moreover, this study, for the first time, adapts the methodology of tool for projecting age patterns using linear splines method (TOPALS) in the Bayesian framework for neighborhoods (instead of municipalities) and enables to account for the similarity in mortality profiles in the neighboring areas. We believe that this paper, in addition to providing the local knowledge on disparities in life expectancy, also provides guidance on the methodological front for future studies seeking to estimate life expectancy in small areas.

In addition to the above, in response to the reviewer's comment, we have substantially revised the abstract, the introduction, and the discussion to highlight the added value of these analyses for understanding urban health inequities generally (beyond the interest specifically in Córdoba).

Here below specific pages and questions that I had while reading your text (I repeat your text as “ and my comments followed by a -)

Comment 1.3

“P90-P10 gap”

- What is p10 or p90? Clarify in the abstract or leave out

Not clear what are those

Response: ‘P90-P10 gap’ refers to the difference between 90th and the 10th percentile of the distribution of life expectancy at birth across small areas, as described in the statistical analysis subsection. We agree with the reviewer on the need to clarify this in the abstract. It now reads:

“We summarized variability in life expectancy at birth by using the difference between 90th and the 10th percentile of the distribution of life expectancy at birth across small areas (P90-P10 gap)...”

Comment 1.4

“mean differences in life expectancy in years per SD higher value...”

- Not clear what is that calculation for?

Response: Thank you for noting this. We have now re-estimated the association between life expectancy and small-area characteristics by calculating a decile-based Slope Index of Inequality (SII) comparing the difference in the life expectancy in the small areas with the highest versus lowest deciles of the socioeconomic variables (see Moreno-Betancur et al., 2015; <https://pubmed.ncbi.nlm.nih.gov/26000548/>). We opted to use this parametrization, instead of using standard deviations (SD) of the predictors, as it is an easier one to explain. Moreover, by using a decile-based SII, this new parametrization is more robust to possible extreme values of predictors in some small areas and allows us to report the estimated regression coefficients in terms of the SII (years). This has been updated in the revised manuscript.

Specifically to the reviewer’s comment, we have added the following clarification to the abstract:

“..., and evaluated the association with small-area socioeconomic characteristics by calculating a Slope Index of Inequality in linear regression.”

Comment 1.5

“acroos”

Response: This typo has been corrected.

Comment 1.6

“These studies are primarily focused on the city of Buenos Aires, but no studies of which we are aware have so far addressed variability and the social patterning of life expectancy at birth in Córdoba, the second most populated city in the country.”

- Why should this be of interest to an international audience? I found this explanation on why looking at Córdoba rather poor. More justification is needed. I would think that comparing BA vs Córdoba as a within country comparison would be much more interesting and appealing for the general reader of BMJ Open.

Response: See response to Comment 1.2. Introduction and discussion have been substantially revised to address this important question.

Comment 1.7

“1) evaluated the variability in life expectancy at birth

- Why not the variability at age at death? Which is actually something of current interest among researchers of mortality

Response: Thank you for this suggestion. Given the lack of evidence on life expectancy, we wanted to first characterize its variability –the widely used and easily understandable metric of overall health– within the city of Córdoba and its association with small-area characteristics, but we agree future studies should examine this issue. We have added the following text to our manuscript:

“ We explored life expectancy as it is a commonly used metric but other metrics such as age at death may also be of value to examine.”

Comment 1.8

**“We hypothesized that (1) there is considerable variability of life expectancy at birth across small areas of Córdoba; (2) there is lower life expectancy in the periphery of the city of Córdoba as compared to core areas; and
- Not enough justification of why is this important beyond for those researchers studying Córdoba**

Response: Please see our response to Comment 1.2.

Comment 1.9

**“3) small areas with better socioeconomic characteristics have higher life expectancy at birth.
- That seems obvious, the rich are living longer all over the world, or what is the research question?**

Response: We understand this concern. As we noted in our response to Comment 1.2 (see Sudharsanan et al., 2020), socioeconomic gradients are not universal, and several Latin American countries have heterogenous social gradients with respect to mortality. As part of our examination of this hypothesis, we assessed not only the heterogeneity in life expectancy and the characterization of its spatial distribution within the city, but also quantified the magnitude of the differences (gaps) between socially distant small areas in the city, given the evidence of large intraurban disparities in large cities of Latin America (Bilal et al., 2019) as noted in our objectives.

Comment 1.10

**“death registries geocoded to small areas during the years 2015-2018
- Details about this death information is important to understand if the allocation of the deaths actually correspond to the living place or the area where hospitals are found**

Response: We have added the following text to the methods:

“Decedents’ residential addresses were extracted from death certificates provided by the Civil Registry of the City of Córdoba and were georeferenced to latitude and longitude coordinates using the ESRI World Geocoding Service by the SALURBAL team.”

Comment 1.11

**“geocoded
- What’s that and how was it done? The place of usual living for the deaths?**

Response: Thank you for noting this. Geocode is mostly used in the US, while the rest of the English-speaking world typically uses georeferencing, so we replaced the term geocoded by georeferenced. See response to Comment 1.10.

Comment 1.12

“we we

Response: This typo has been corrected.

Comment 1.13

**“Second, we obtained population projections
- From where?”**

Response: We clarified this in the manuscript (see Population Data subsection). The sentence now states:

“...we obtained projected population estimates provided by the National Institute of Statistics and Census of Argentina for 2015-2018 by 5-year age groups (0 to 85+ years) and sex for the whole municipality of Cordoba, ...”

Comment 1.14

**“We used all mortality records for the city of Cordoba from 2015 to 2018
- Where is the information coming from?”**

Response: Thank you for noting this. See response to Comment 1.10 .

Comment 1.15

**“socioeconomic characteristics by small area from the 2010 national population census
- It is extremely inconvenient that the timing of the data is not matching.”**

Response: We are aware of this limitation, and we have explicitly mentioned in the limitation section that we don't have small-area socioeconomic characteristics data from 2015-2018. As highlighted in the text, we assumed that the social characteristics of the small areas (from 2010) were relatively stable until the years of death examined (2015-2018). Even though neighborhoods are dynamic units and can experience modifications, it is unlikely that the features of small areas changed substantially during the relatively short window of years we examined. Additionally, we think that the usage of a combined socioeconomic indicator that incorporates information on several social characteristics at once may help minimize potential variations in a particular socioeconomic feature during that time. We added the following text to the limitations section:

“While ideally this analysis would have used census data from other years to better assess small-area changes over time, we could only rely on the 2010 census because small-area data from the 2001 Argentina census were not available, and the 2020 census was postponed to 2022 due to the COVID-19 pandemic, and data are not available yet.”

Comment 1.16

**“standard mortality schedule were noisy
- Noisy? Doesn't sound as a way of describing rates in a scientific journal**

Response: We clarified this in the text. The sentence now reads:

“Because the log mortality rates of the standard mortality schedule were noisy (i.e., contained unexplained variance) (see Supplementary material, Figure S1), we fit a LOESS regression, and the resulting smoothed rates were used as the standard mortality schedule.”

Comment 1.17

**“calling the software
“discarded as burn-ins
“samples were thinned
- A lot of jargon. Doesn't sound as usual wording for a methods section of a manuscript.**

Response: The terms “burn-ins” or “thinning of samples” are commonly used in Bayesian statistics (see: Quick et al., 2020; <https://pubmed.ncbi.nlm.nih.gov/31688128/>). Anyhow, we agree with the reviewer and have now rephrased the term “Calling the software”. The sentences now states:

“The models were run in WinBUGS[27] by using the R package R2WinBUGS[28] for 100,000 iterations and the first 80,000 samples were discarded as burn-ins. The remaining samples were thinned by a factor of 10 to reduce the autocorrelation of the samples.[22].”

Comment 1.18

“To estimate the amount of variability in life expectancy at birth for each sex (hypothesis 1), we calculated the difference between 90th and the 10th percentile of the distribution of life expectancy at birth (P90-P10 gap)

- that needs to be clarified in the abstract

Response: See, response to Comment 1.3. The sentence now states:

“We summarized variability in life expectancy at birth by using the difference between 90th and the 10th percentile of the distribution of life expectancy at birth across small areas (P90-P10 gap)...”

Comment 1.19

“A median of 222 and 199 deaths occurred during the study period per small area among women and men, respectively (Table 1).

- Those are very small numbers of deaths which if proper confidence intervals had been calculated will translate in very big CIs. Then how to compare the P90 to the P10 and their CIs, for example if the CIs overlap. Not clear.

Response: Thank you for sharing this concern. We agree that in classical approaches, these numbers could be a cause for concern with estimating life expectancy. As noted in our manuscript, the Bayesian random-effects approach for the estimation of life expectancy (and its confidence intervals), is appropriate and feasible even with small numbers of events (see Jonker et al., 2012: <https://academic.oup.com/aje/article/176/10/929/92314>). In terms of confidence intervals, the P90-P10 gaps are not related to confidence intervals, rather they represent the variability in life expectancy across the city, not within each area. We have better described this in the text now:

“To estimate the amount of variability in life expectancy at birth for each sex (hypothesis 1), we calculated the difference between 90th and the 10th percentile of the distribution of life expectancy at birth (P90-P10 gap) across small areas. The P90-P10 gap represents the variability in life expectancy across the city.”

We also note the challenges of small data in our discussion. The text now reads:

“A key strength of our analyses is that we employed a novel Bayesian approach that draws strength and smooths estimates from surrounding small areas and from the overall structure of the mortality schedule. This allowed us to derive valid and reliable estimates for small areas even in the presence of sparse data”.

Comment 1.20

“The distribution of life expectancy at ages 20, 40, and 60 years, showed similar patterns to those for life expectancy at birth in women and men (see Figure S3, Supplementary material).

- Life expectancy by age is practically a straight line and you are looking at parallel lines when comparing the different small areas in the city.

Response: We agree that life expectancies at different ages are strongly correlated; nevertheless, by examining life expectancy at different ages, we can explore what ages are (and more importantly, are

not) driving these differences. For example, we would expect to find a different spatial distribution of life expectancy at age 20 if infant mortality had a considerable impact on the estimation of life expectancy at birth, and different distributions might arise when mapping life expectancy at 40 if violent deaths (that peak at 15-39) are important in life expectancy at birth. The sentence now states:

“We also extracted life expectancy at ages 20, 40, and 60 years from life tables to explore what ages could drive differences in life expectancy at birth.”

Also, we added the term *spatial* in the following sentence:

“The distribution of life expectancy at ages 20, 40, and 60 years, showed similar spatial patterns”.

Comment 1.21

“1-SD

- Not clear why first you show some results with P90-P10 and then variability using 1-SD.

Response: Originally, we calculated the gap P90-P10 when evaluating the variability in life expectancy across small areas (hypothesis 1), but 1-SD only when estimating the associations between life expectancy with characteristics of small areas (hypothesis 3). As described in the response to Comment 1.4, we have updated the parameterization, replacing SD-based parameterization with a decile-based Slope Index of Inequality (SII). This compares the difference in life expectancy in the highest vs. lowest deciles of the small-area socioeconomic characteristics, providing results that are easier to communicate to a general audience. This has been updated in the Statistical analysis subsection, as follows:

“Finally, to examine the association between life expectancy at birth and socioeconomic characteristics of small areas (hypothesis 3), we fit univariate linear regressions of life expectancy on each predictor variable converted into deciles and scored on a continuous scale between 0 to 1. Specifically, for each socioeconomic variable, we assigned the value of 0 if it corresponds to the first decile of its distribution across all small areas. The second decile got the value of 1/9, the third decile got the value of 2/9, and so on.”

“Each coefficient represents the mean difference in life expectancy in areas with the highest socioeconomic variable (i.e., those in the tenth decile) versus the areas with the lowest value of the socioeconomic variable (those in the first decile), and is presented as the Slope Index of Inequality (SII).”

Comment 1.22

“Argentina has been mainly focused in the city of Buenos Aires.[4,26]

- Again, then why not a comparison between the main cities of Argentina

Response: Please see the response to Comment 1.2.

Comment 1.23

“Their distribution showed similar patterns to those reported for life expectancy at birth, suggesting that our estimates of life expectancy at birth are reliable and robust to those factors.

- The strong relation between mortality levels at different ages is at the core of empirical model life tables. So your finding that results match across ages is not a reflection of better or worse estimation it is simply a result of this strong mortality relations at different ages.

Response: Thank you for noting this, we agree with this assessment and have removed that sentence.

Comment 1.24

“We also had to combine population projections

- What do you mean by combining population projections? And do you mean population projections? That is usually into the future

Response: We have added clarity to this sentence in the subsection Data source, Population data (Data source):

“Second, we obtained projected population estimates provided by the National Institute of Statistics and Census of Argentina for 2015-2018 by 5-year age groups (0 to 85+ years) and sex for the whole municipality of Córdoba and graduated this data into single ages using a penalized composite link model (PCLM).”

In the Discussion under Limitations, we changed this phrasing to:

“Because annual small-area population by age and sex were not available, we had to estimate these by combining official municipality-level projected population estimates from 2015-2018 with the most recent census in 2010.”

Comment 1.25

“If the population age or sex structure changed over time differentially by area, our estimates may be biased

- This mismatching in the timing of the information should be somehow incorporated in the regression analysis. For example, different census (2000 and 2010, 2020) could be used to actually confirm such fixed values or somehow do interpolation of the information.

Response: We appreciate this suggestion. Unfortunately, 2010 is the only census with small area data available that we were able to obtain. The 2020 census was postponed to May 2022 due to the COVID-19 pandemic and the data are not available yet. The 2001 census data are publicly available only at the municipality level and we were unable to obtain the small-area level data. We added this limitation at the end of the discussion, as follows:

“While ideally this analysis would have used census data from other years to better assess small-area changes over time, we could only rely on the 2010 census because small-area data from the 2001 Argentina census were not available, and the 2020 census was postponed to 2022 due to the COVID-19 pandemic, and data are not available yet.”

Comment 1.26

Figure 1

- How is density included in the calculations of life expectancy?

Response: We are not sure about what is meant by ‘density’. Information like population density, etc., are not included anywhere in estimating the mortality rate, and by extension, life expectancy. Figure 1 is a choropleth map showing categories of life expectancy by sex across small areas in the city and does not reflect densities of any type.

Comment 1.27

Figure 2

- Strange, confidence intervals do not cover the values, and does not align to them. In other words, I would expect that if there was greater number of observed dots with greater disparity that the CI will cover the great majority... that is not the case, instead giving wide CI when single dots are found

Response: The discrepancy could be related to the fact that what we have shown are confidence intervals around the mean but not the prediction interval, which by definition is much wider because it

has to account for uncertainty in estimating each data point. To facilitate the interpretation, we decided to remove the CI's. The footnote of Figure 2 reads:

“Figure 2. Association between life expectancy at birth and socioeconomic characteristics of small areas in women and men. Córdoba, 2015-2018. Footnote: Blue line refers to the linear fit.”

Comment 1.28

- What are the yellow and blue lines within CI?

Response: Thank you for noting this. Actually, this was originally added to the legend of Figure 2, which was included in a separate section during the submission process; that is why this information was missing in Figure 2. The blue line referred to the linear fit (95% CI) and the yellow one to the LOESS fit. However, we have decided to remove the LOESS fit in order to make the plots easier to interpret.

Comment 1.29

Figure S1

- Why do you need to smooth them? Justification is needed

Response: We have now added a clarification in the footnote of Figure S1; see response to Comment 1.16.

Comment 1.30

Figure S2

- What is the vertical axis? Label?

Response: The vertical axis corresponds to the small areas. We have now added a footnote to Figure S2 for clarification, as follows:

“Footnote: small areas are represented in the vertical axis.”

REVIEWER 2

Lucia Andreozzi

Comments to the Author:

I consent to the publication of this review, it is clear, well written and it uses statistical models properly.

Response: Thank you for your comments on the manuscript.

References

Bilal U, Hessel P, Perez-Ferrer C, *et al.* Life expectancy and mortality in 363 cities of Latin America. *Nat Med* 2021;**27**:463-470.

Bilal U, Alazraqui M, Caiaffa WT, *et al.* Inequalities in life expectancy in six large Latin American cities from the SALURBAL study: an ecological analysis. *Lancet Planet Health* 2019;**3**:e503–10.

Jonker MF, van Lenthe FJ, Congdon PD, *et al.* Comparison of Bayesian Random-Effects and Traditional Life Expectancy Estimations in Small-Area Applications. *Am J Epidemiol* 2012; **176**(10):929-937.

Kii M. Projecting future populations of urban agglomerations around the world and through the 21st century. *npj Urb Sustain.* 2021;**1**:1–12.

Moreno-Betancur M, Latouche A, Menvielle G, *et al.* Relative index of inequality and slope index of inequality: a structured regression framework for estimation. *Epidemiology* 2015;**26**(4):518-27.

Sudharsanan N, Zhang Y, Payne CF, *et al.* Education and adult mortality in middle-income countries: Surprising gradients in six nationally-representative longitudinal surveys. *SSM Popul Health* 2020;**12**:100649.

Quick H, Terloyeva D, Wu Y, *et al.* Trends in Tract-Level Prevalence of Obesity in Philadelphia by Race-Ethnicity, Space, and Time. *Epidemiology* 2020;**31**(1):15-21.

VERSION 2 – REVIEW

REVIEWER	Canudas-Romo, Vladimir Australian National University, School of Demography, College of Arts and Social Sciences
REVIEW RETURNED	16-Aug-2022
GENERAL COMMENTS	Thank you for taking good care of all my comments. I am still concerned that the data is not fully open to all to use. For example, aggregated information as that in the Human Mortality Database could be made available without compromising any identifiability iss